# Weakly-Supervised Disentangled Representation Learning via Filter-Based Adaptive Swapping

**Zhenyu Zong**                                                          *zzong@wm.edu*
*Department of Computer Science*
*William & Mary*

**Qidi Wang**                                                          *qwang25@wm.edu*
*Department of Computer Science*
*William & Mary*

**Simon Yu**                                                          *jundayu2@illinois.edu*
*Department of Electrical and Computer Engineering*
*University of Illinois Urbana-Champaign*

**Hongpeng Cao**                                                          *cao.hongpeng@tum.de*
*School of Engineering and Design*
*Technical University of Munich*

**Yanbing Mao**                                                          *hm9062@wayne.edu*
*Engineering Technology*
*Wayne State University*

**Han Zhao**                                                          *hanzhao@illinois.edu*
*Department of Computer Science*
*University of Illinois Urbana-Champaign*

**Lui Sha**                                                          *lrs@illinois.edu*
*Department of Computer Science*
*University of Illinois Urbana-Champaign*

**Huajie Shao**                                                          *hshao@wm.edu*
*Department of Computer Science*
*William & Mary*

**Reviewed on OpenReview:** *https://openreview.net/forum?id=K69rKKozZU*

## Abstract

Disentangled representation learning (DRL) aims to uncover semantically meaningful latent factors from observed data, thereby improving both interpretability and generalization of machine learning (ML) models. Despite remarkable progress, unsupervised DRL cannot achieve complete disentanglement without inductive biases or supervision. To address this challenge, existing approaches either rely on full supervision, which demands extensive manual labeling, or weak supervision, which involves complex training strategies that often result in unstable training. To address these limitations, we propose Filter-VAE, a weakly supervised variational autoencoder (VAE) that introduces a filter-based adaptive swapping strategy to learn stable and meaningful disentangled representations. Specifically, a relevance filter removes semantically meaningless latent factors, while an adaptive swapping filter exchanges those latent factors that have reached stability. With these two filters, Filter-VAE adaptively swaps only stable and semantically aligned latent factors, leading to robust and meaningful representations. We evaluate Filter-VAE on three

standard benchmarks and our created traffic sign dataset in two downstream tasks: disentanglement and adversarial robustness. Experimental results demonstrate that Filter-VAE achieves strong disentanglement performance with reduced supervision and delivers remarkable robustness against diverse adversarial attacks and corruptions. The code is released at `https://github.com/ZY-Zong/Filter-VAE.git`.

# 1 Introduction

Disentangled representation learning (DRL) aims to uncover semantically meaningful latent representations that independently capture the underlying generative sources of variation in observed data. Such representations contain high-level concepts that are understandable to humans—for example, the shape and color of different traffic signs. This offers several benefits, including improved interpretability, enhanced sample efficiency in downstream tasks, and better generalization capability across domains. DRL has demonstrated success in several domains such as image generation, natural language processing, and recommendation systems (Wang et al., 2024).

Over the past few years, a wide range of DRL methods have been proposed, broadly categorized into three groups: unsupervised, fully supervised, and weakly supervised approaches. Unsupervised methods such as $\beta$-VAE (Higgins et al., 2017) and FactorVAE (Kim & Mnih, 2018) aim to discover disentangled factors without any labeled supervision. Despite the success of unsupervised DRL on synthetic datasets, Locatello et al. (2019) have pointed out that unsupervised methods are not effective without inductive biases or auxiliary information. To address this issue, some works have introduced fully-supervised DRLs, such as Concept Bottleneck Models (Koh et al., 2020) and ML-VAE (Bouchacourt et al., 2018), which use explicit annotations that are aligned with latent variables to guide the learning process toward meaningful latent factors. However, existing methods are limited by the need for extensive supervision, which may be costly and impractical in many real-world applications.

Weakly-supervised approaches offer a compromise by leveraging limited or indirect supervision, such as partial labels, pairwise similarity, causal priors, and labeling functions (Feng et al., 2018; Locatello et al., 2020a; Shen et al., 2022; Tonolini et al., 2023), combined with specifically designed training strategies. For example, a very recent method, SW-VAE (Zhu et al., 2023), utilizes annotations indicating the maximum number of varying factors and introduces an optimized latent swapping strategy to improve disentanglement. However, SW-VAE still faces two major limitations. First, it relies on a warm-up stage that swaps latent factors without confirming whether semantically meaningful features have been learned. This can lead to unstable training when the latent factors are not yet fully disentangled. Second, SW-VAE requires annotations specifying multiple distinct generative factors for all pairwise samples, which still demands substantial labeling effort. More recent weakly-supervised methods have shifted their focus toward other downstream tasks such as binary classification (Tonolini et al., 2023), image restoration (Zheng et al., 2024), and 3D face modeling (Li et al., 2024), rather than disentanglement.

Motivated by the limitations of SW-VAE, we propose Filter-VAE, a weakly-supervised variational autoencoder that employs a filter-based adaptive swapping strategy to learn stable and meaningful disentangled representations. However, developing Filter-VAE poses two key challenges: (i) not all latent dimensions encode semantically meaningful features, and (ii) frequently swapping latent factors that are not yet fully disentangled can result in unstable feature learning.

To address the first challenge, we propose a relevance filter that removes meaningless (noisy) latent factors by applying a threshold to identify meaningful latent factors. To tackle the second challenge, we introduce an adaptive swapping filter that identifies stable latent factors by computing the KL divergence between pairs of latent variables and then adaptively swaps them based on this divergence. Compared to previous works, the combination of these two filters can locate positions of semantically meaningful latent factors and prevent the swapping of unstable latent features. Additionally, our method requires less prior knowledge, needing only a single distinct generative factor for a small amount of the pairwise data. We evaluate Filter-VAE on multiple benchmark datasets and a created traffic sign dataset in two downstream tasks: disentanglement and adversarial robustness. Extensive experimental results demonstrate that our method

outperforms both unsupervised and weakly-supervised baselines in the disentanglement task. Moreover, it achieves superior adversarial robustness compared to existing defense baselines under various attacks and corruptions. These findings underscore the effectiveness of the proposed method in learning robust and stable latent representations from observed data.

**In summary, our contributions** are three-fold as follows:

- **Novel disentanglement learning model**. We propose Filter-VAE, a novel weakly supervised VAE that incorporates two filters to achieve stable disentanglement using a small amount of labeled pairwise data.

- **Synthetic traffic sign dataset**. We create a new synthetic traffic sign dataset to assess both the disentanglement performance and adversarial robustness of our method.

- **Superior performance in two downstream tasks**. Evaluation results demonstrate that the proposed Filter-VAE achieves better disentanglement performance than the baselines on multiple datasets with less supervision. Moreover, it can significantly enhance the adversarial robustness of ML predictions under 17 adversarial attacks and corruptions.

## 2 Related Work

### 2.1 Unsupervised DRL

One representative type of unsupervised DRL is the Variational Auto-encoder (VAE) (Kingma & Welling, 2014), which encompasses an encoder and a decoder. The core idea is to model data distributions by maximizing their variational inference in an unsupervised manner (Wang et al., 2024). In the past few years, many studies have proposed unsupervised VAE and its variants (Burgess et al., 2018; Chen et al., 2018), such as $\beta$-VAE (Higgins et al., 2017), FactorVAE (Kim & Mnih, 2018), and ControlVAE (Shao et al., 2020b; 2022), to learn the disentangled representations from observed data without the requirement of labels. These methods introduce the trade-off between reconstruction quality and disentanglement by adding a new regularizer in the objective function, such as mutual information or total correlation (Chen et al., 2018). While they have shown good performance in synthetic datasets, Locatello et al. (2019) pointed out that it is theoretically impossible to fully disentangle latent factors using unsupervised learning without inductive bias.

### 2.2 Fully-supervised DRL

Fully-supervised DRL approaches address the limitation of unsupervised methods by leveraging full supervision, where ground-truth generative factors or concept annotations are assumed available. Models like Label Supervised VAE (Bouchacourt et al., 2018), directly conditioned on labeled factors, achieving disentanglement by aligning latent variables with supervised attributes. Concept Bottleneck Models (CBMs) (Koh et al., 2020) and ProtoPNet (Chen et al., 2019), focused on interpretable classification by forcing the model to base decisions on human-defined concepts. CausalVAE (Yang et al., 2021) further leveraged supervision signals to learn structural causal model as prior. While supervised DRL can achieve high disentanglement quality, it relies heavily on labeled data, which is often costly or infeasible to obtain in real-world scenarios.

### 2.3 Weakly-supervised DRL

To avoid massive labeling requirements while still achieving disentanglement, researchers have explored weakly-supervised VAEs that leverage partial or indirect supervision. Early approaches leverage limited labels on input subsets (Locatello et al., 2020b) or pairwise annotations (Locatello et al., 2020a). Subsequent works extend this ides using different forms of supervision such as similarity measurements (Chen & Batmanghelich, 2020), the number of different generative factors (Locatello et al., 2020a), and the maximum number of different generative factors (Zhu et al., 2023). More recently, weakly-supervised VAEs

have been adapted for other downstream tasks, including binary classification (Tonolini et al., 2023), image restoration (Zheng et al., 2024), and 3D face modeling (Li et al., 2024), which lie outside the scope of disentanglement itself.

A parallel line of work builds on the idea of latent swapping, which is first introduced by DNA-GAN (Xiao et al., 2018) in the supervised setting. This strategy has been adopted in both self-supervised and weakly-supervised VAEs. For example, Swap-VAE (Liu et al., 2021) proposed block-wise swapping in self-supervised framework, DSD (Feng et al., 2018) extended it with dual swapping under weakly supervision, and SW-VAE (Zhu et al., 2023) further optimized it by progressively increasing the number of swapping factors. However, these methods often perform swaps without verifying whether the swapped factors are semantically meaningful, leading to unstable training if the latent factors are not yet fully disentangled. In this work, we address these shortcomings by introducing an adaptive swapping strategy that delays swapping until semantically meaningful latent factors have stabilized. Moreover, our method achieves disentanglement with less supervision, requiring only 10% of pairwise inputs annotated with a single distinct generative factor.

## 3 Preliminaries

### 3.1 Problem Statement

The goal of this work is to learn semantically meaningful latent representations from observations with limited supervised signals. The primary challenge lies in effectively disentangling independent latent factors that are aligned with the ground truth generative factors. Given a batch of $N$ observed image $\boldsymbol{x} = \{\boldsymbol{x}_1, \boldsymbol{x}_2, \ldots, \boldsymbol{x}_N\}$, DRL aims to find a mapping $f(\boldsymbol{x}) \rightarrow \boldsymbol{z}$, where $\boldsymbol{z} \in \mathbb{R}^d$ denotes the learned factors in a latent space with the dimension size of $d$ (Eastwood & Williams, 2018). Each *latent factor* is expected to represent one and only one *explanatory factor* of $\boldsymbol{x}$ so that modifying one latent factor doesn't affect the other factors of variation. This guarantees each latent factor is independent of the others. However, since some factors are not semantically meaningful, the number of explanatory factors $\boldsymbol{v}$ is always less than or equal to the number of latent factors; namely, $|\boldsymbol{v}| \leq d$.

### 3.2 Variational Autoencoder

Variational autoencoder (VAE) (Kingma & Welling, 2014) is a popular unsupervised deep generative model. It consists of two main components: (1) The encoder $q_\phi(\boldsymbol{z}|\boldsymbol{x})$ maps input data $\boldsymbol{x}$ into a probabilistic distribution in the latent space, which is often assumed to be a Unit Gaussian. (2) The decoder $p_\theta(\boldsymbol{x}|\boldsymbol{z})$ reconstructs the input based on the representations sampled from the latent space. In general, researchers often optimize the following evidence lower bound (ELBO):

$$\mathcal{L}_{vae} = \mathbb{E}_{q_\phi(\boldsymbol{z}|\boldsymbol{x})}[\log p_\theta(\boldsymbol{x}|\boldsymbol{z})] - D_{KL}(q_\phi(\boldsymbol{z}|\boldsymbol{x})\|p(\boldsymbol{z})). \tag{1}$$

To better learn disentangled representations, variants like $\beta$-VAE (Higgins et al., 2017) are proposed, which adds $\beta$ ($\beta > 1$) in front of the KL-divergence in the objective. However, a large and fixed $\beta$ could lead to poor reconstruction quality (Shao et al., 2020a).

### 3.3 ControlVAE

To effectively trade off the reconstruction quality and disentanglement performance, ControlVAE (Shao et al., 2020b) incorporates a controller from automatic control into the basic VAE model to dynamically adjust the hyperparameter $\beta$ during training. The objective of ControlVAE is given by

$$\mathcal{L}_{control}(\boldsymbol{x}, \boldsymbol{z}) = \mathbb{E}_{q_\phi(\boldsymbol{z}|\boldsymbol{x})}[\log p_\theta(\boldsymbol{x}|\boldsymbol{z})] - \beta(t) D_{KL}(q_\phi(\boldsymbol{z}|\boldsymbol{x})\|p(\boldsymbol{z})), \tag{2}$$

where $\beta(t)$ is the output of a non-linear PI controller as follows:

$$\beta(t) = \frac{K_p}{1 + \exp(e(t))} - K_i \sum_{j=0}^{t} e(j) + \beta_{min}, \tag{3}$$

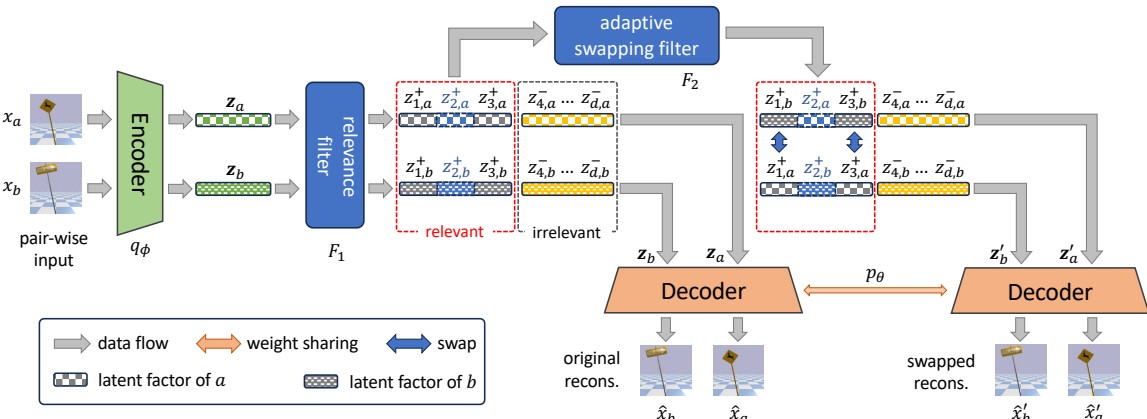

Figure 1: The overall framework of weakly-supervised Filter-VAE. A pair of images with one distinct generative factor is fed into the encoder $q_\phi$ to generate latent factors. Then two filters are implemented to guarantee stable swapping: (1) relevance filter $F_1$ filters out irrelevant (meaningless) latent factors $\boldsymbol{z}^-$ for only swapping relevant (meaningful) factors $\boldsymbol{z}^+$ in the latent space; (2) Adaptive swapping filter $F_2$ sets an upper bound for KL-divergence to prevent swapping undesired latent factors $(z_2^+)$ and unstable latent factor during training. It then swaps all stable and relevant latent factors $(z_1^+, z_3^+)$ except for the most different one $(z_2^+)$. The decoder reconstructs the input data based on both original $(\boldsymbol{z}_a, \boldsymbol{z}_b)$ and swapped $(\boldsymbol{z}_a', \boldsymbol{z}_b')$ latent factors.

where $K_p$ and $K_i$ are hyperparameters of the PI algorithm, and $\beta_{min}$ is an application-specific constant. $e(t)$ is the error between the designed KL-divergence and the real output.

Like other unsupervised VAE-based methods, ControlVAE still struggles to achieve perfect disentanglement from complex data. To address this problem, we will extend ControlVAE to develop a novel weakly-supervised model for disentangled representation learning.

## 4 Proposed Method

As mentioned above, current weakly-supervised DRL methods, such as SW-VAE (Zhu et al., 2023), perform latent factor swapping without accounting for the disentanglement status of the latent space, often resulting in training instability. To solve the limitation, we propose a new weakly-supervised Filter-VAE in this work. As shown in Figure 1, Filter-VAE consists of four main components: encoder, relevance filter, adaptive swapping filter, and decoder, *with two filters as cores*. The main idea is that it first encodes the input pairwise data into latent representations. The relevance filter then filters out meaningless latent factors. After that, the adaptive swapping filter finds disentangled stable factors and implements swapping. Finally, the swapped latent features are decoded to reconstruct the input data. We elaborate on each component in the following section.

### 4.1 Encoder for latent representation learning

First, we adopt an encoder $q_\phi$ to learn latent representations of pairwise input. Let $(\boldsymbol{x}_a, \boldsymbol{x}_b)$ denote a pair of samples with one distinct factor. As shown in Figure 1, the encoder $\boldsymbol{q}_\phi$ maps $(\boldsymbol{x}_a, \boldsymbol{x}_b)$ into latent variables $(\boldsymbol{z}_a, \boldsymbol{z}_b)$, respectively. Latent variables are expressed as probabilistic Gaussian distributions, which are approximated by a convolutional neural network. The encoded representations can be formulated as:

$$q_\phi(\boldsymbol{z}_i|\boldsymbol{x}_i) \sim \mathcal{N}(\mu_{\boldsymbol{z}_i}, \mathrm{diag}(\sigma_{\boldsymbol{z}_i}^2)), \tag{4}$$

where $i \in \{a, b\}$. $\mu_{\boldsymbol{z}_i}$ and $\sigma_{\boldsymbol{z}_i}^2$ represents the mean and variance of latent variable $\boldsymbol{z}_i$. Latent variables are obtained with the reparameterization trick to allow tractable computation (Kingma & Welling, 2014). $\boldsymbol{z}_i$ contains $d$ latent factors $\{z_{1,i}, z_{2,i}, \ldots, z_{d,i}\}$, where not all of them are semantically meaningful.

## 4.2 Relevance filter

We then use the relevance filter to remove irrelevant latent factors, thereby overcoming the current methods' limitation that often swap some meaningless latent factors during model training. To achieve this, we propose to separate all latent factors into relevant and irrelevant parts. Inspired by empirical experiments in prior work (Shao et al., 2020b), we can see that meaningful factors exhibit higher dim-wise KL divergence than those meaningless ones. Based on this observation, we develop a filter $F_1$ with a threshold $\alpha$ to select meaningful factors below:

$$F_1(\alpha, z_i) = \begin{cases} z_i^+, & \text{if} \quad D_{KL}(z_i) \geq \alpha \\ z_i^-, & \text{if} \quad D_{KL}(z_i) < \alpha \end{cases}. \tag{5}$$

Here, suppose the KL divergence between the $i$-th latent factor of a pair of data samples exceeds the threshold $\alpha$. In this case, we consider this latent factor to encompass semantically meaningful concepts, such as shapes and colors. These latent factors are relevant for expressing meaningful high-level features of the data and are denoted with a + symbol.

## 4.3 Adaptive swapping filter

After identifying relevant latent factors, the third step is to use the second filter to find stable latent factors for implementing swapping. Since latent factors in early training iterations may not be disentangled well, swapping these factors may result in training instability. To overcome this issue, we design the second filter $F_2$ based on dim-wise KL-divergence between $\boldsymbol{z}_a^+$ and $\boldsymbol{z}_b^+$ as follows:

$$\begin{aligned} D_{KL}\left(q_{\phi,a}(z_i^+)\|q_{\phi,b}(z_i^+)\right) &= \int q_{\phi,a}(z_i^+) \log \frac{q_{\phi,a}(z_i^+)}{q_{\phi,b}(z_i^+)} dz_i^+ \\ &= -\frac{1}{2}\left(\log \frac{\sigma_{i,a}^2}{\sigma_{i,b}^2} - \frac{\sigma_{i,a}^2 + (\mu_{i,a} - \mu_{i,b})^2}{\sigma_{i,b}^2} + 1\right), \end{aligned} \tag{6}$$

where $q_{\phi,a}(z_i^+)$ and $q_{\phi,b}(z_i^+)$ denote the posterior probability of the $i$-th latent factor learned by the encoder $\boldsymbol{q}_\phi$. Both of them follow Gaussian distribution $\mathcal{N}(\mu_i, \sigma_i^2)$ (Burgess et al., 2018). For simplification, we use $D(z_i^+)$ to represent $D_{KL}\left(q_{\phi,a}(z_i^+)\|q_{\phi,b}(z_i^+)\right)$ in the following.

Next, we need to figure out which latent factor should be swapped based on Eq. equation 6. A straightforward method is to swap all meaningful factors except for the one with the largest dim-wise KL-divergence, denoted by $j = \text{argmax}_i D(z_i^+)$. This seems effective because there exists only one distinct factor for each pairwise input sample. However, in practice, we find that sometimes the largest $D(z_i^+)$ doesn't represent the most distinct factor between a pair of samples during model training if the latent distribution is not learned well, especially in early training iterations. As a result, swapping could occur unexpectedly in the non-target factors, leading to training instability. To resolve this problem, we further set an upper bound $\gamma$ for $D(z_i^+)$ to avoid unexpected swapping. Thus, we have the second filter $F_2$ to determine which latent factor should be swapped:

$$F_2(\gamma, z_i^+) = \begin{cases} \text{swap}^- \text{ if } \max D(z_i^+) > \gamma \\ \text{swap}^- \text{ if } \begin{array}{l} \max D(z_i^+) \leq \gamma \\ \text{and } i = \text{argmax}_k D(z_k^+) \end{array} \\ \text{swap}^+ \text{ if } \begin{array}{l} \max D(z_i^+) \leq \gamma \\ \text{and } i \neq \text{argmax}_k D(z_k^+) \end{array} \end{cases} \tag{7}$$

where $\gamma$ is a threshold, swap$^+$ means swapping will be implemented and swap$^-$ means not. The first condition indicates that we will not implement unexpected swapping when $D(z_i^+)$ is greater than $\gamma$. This can avoid swapping entangled latent factors. For the second condition that $D(z_i^+)$ is lower than $\gamma$, if the $i$-th latent factor corresponds to the largest dim-wise KL divergence, we do not implement swapping since it is more likely to be a distinct factor. If the first and second conditions do not hold, we will swap the latent factors.

### 4.4 Decoder for reconstruction

After swapping latent factors, the final step is reconstructing the input data. We input both the original latent factors $z_a$, $z_b$ and swapped factors $z_a{}'$, $z_b{}'$ into the decoder $p_\theta$ to reconstruct input data $\hat{x}_a$, $\hat{x}_b$, $\hat{x}_a'$, and $\hat{x}_b'$, respectively. Both reconstructions should remain identical to their original inputs because the swapping occurs between latent factors with the same semantic meaning. This motivates the design of a reconstruction objective that encourages latent factors to align with the desired distributions.

### 4.5 Objective of DRL

The overall objective of DRL comprises two loss functions: the VAE loss and the reconstruction loss between the original and reconstructed samples. Specifically, we combine the ControlVAE introduced in Sec. 3 with the proposed swapping technique to learn disentangled representations. The loss of ControlVAE for pairwise input samples can be written as

$$\mathcal{L}_{pair} = \mathcal{L}_{control}(\hat{x}_a, z_a) + \mathcal{L}_{control}(\hat{x}_b, z_b). \tag{8}$$

After swapping, we ensure that the reconstructed image is close to the original input. Thus, we have the following reconstruction loss

$$\mathcal{L}_{swap} = \|x_a - \hat{x}_a'\|_2^2 + \|x_b - \hat{x}_b'\|_2^2. \tag{9}$$

Combining the ControlVAE loss and reconstruction loss above, the overall objective is given by

$$\mathcal{L}_{overall} = \mathcal{L}_{pair} + \omega \mathcal{L}_{swap}, \tag{10}$$

where $\omega$ is a hyperparameter to balance the second term.

We summarize the proposed Filter-VAE in Algorithm 1. Lines 5-12 aim to extract meaningful latent factors from the latent representations. In lines 17-21, we swap the identical latent factors between a pair of samples with one distinct factor.

## 5 Experiments

We first evaluate the disentanglement performance of Filter-VAE on three benchmark datasets and one traffic sign dataset that we created. Then we verify its robustness against adversarial attacks in traffic sign detection. Lastly, we explore the impact of important hyperparameters and components on model performance. The detailed model configurations and hyperparameter settings are presented in Appendix A.

### 5.1 Datasets

#### 5.1.1 Benchmark datasets

We first evaluate the disentanglement performance of our method using three benchmark datasets: **dSprites** (Matthey et al., 2017), **3dShapes** (Burgess & Kim, 2018), and **3dChairs** (Aubry et al., 2014). Each benchmark contains different ground-truth factor labels and corresponding numbers of factors:

1. **dSprites** (Matthey et al., 2017) contains $737,280$ binary $64 \times 64$ images of 2D shapes generated by 6 ground truth factors (number of factors): color (1), shape (3), scale (6), orientation (40), x-position (32), y-position (32).

2. **3dShapes** (Burgess & Kim, 2018) contains $480,000$ RGB $64 \times 64 \times 3$ images with 6 ground truth factors (number of factors): floor hue (10), wall hue (10), object hue (10), scale (8), shape (4), orientation (15).

3. **3dChairs** (Aubry et al., 2014) contains $86,366$ synthesized RGB $64 \times 64 \times 3$ images with 3 ground factors (number of factors): style (1,393), horizontal orientation (31), vertical orientation (2).

---

**Algorithm 1:** The Filter-VAE Algorithm

---

**Input:** Pairwise data samples $\{(\boldsymbol{x}_a, \boldsymbol{x}_b)_i, i = 1, 2, \ldots N\}$, latent dimension $d$, threshold $\alpha$ for KL,
            threshold $\gamma$ for adaptive swapping, iterations $T$

**Output:** $\phi_T$, $\theta_T$

**1** Initialize $\phi_1$, $\theta_1$

**2** **for** $t = 1$ **to** $T$ **do**

**3**      Sample a pair of images $(\boldsymbol{x}_a, \boldsymbol{x}_b)$

**4**      $\boldsymbol{z}_a, \boldsymbol{z}_b \leftarrow q_{\phi_t}(\boldsymbol{x}_a), q_{\phi_t}(\boldsymbol{x}_b)$

**5**      **for** $i = 1$ **to** $d$ **do**

**6**          $D_{KL}(z_i) \leftarrow \frac{1}{2}(D_{KL}(z_{i,a}) + D_{KL}(z_{i,b}))$

**7**          **if** $D_{KL}(z_i) \geq \alpha$ **then**

**8**              Obtain relevant factors $z_{i,a}^+, z_{i,b}^+$

**9**          **else**

**10**             Get meaningless factors $z_{i,a}^-, z_{i,b}^-$

**11**          **end**

**12**      **end**

**13**      **for** $i = 1$ **to** $|\boldsymbol{z}^+|$ **do**

**14**          $D(z_i^+) \leftarrow D_{KL}(q_{\phi_t,a}(z_i^+) \| q_{\phi_t,b}(z_i^+))$

**15**      **end**

**16**      $\boldsymbol{z}_a', \boldsymbol{z}_b' \leftarrow \boldsymbol{z}_a, \boldsymbol{z}_b$

**17**      **for** $i = 1$ **to** $|\boldsymbol{z}^+|$ **do**

**18**          **if** $\max D(z_i^+) \leq \gamma$ **and** $i \neq \operatorname{argmax} D(z_i^+)$ **then**

**19**              Swap $z_{i,a}'^+$ and $z_{i,b}'^+$

**20**          **end**

**21**      **end**

**22**      $\hat{\boldsymbol{x}}_a, \hat{\boldsymbol{x}}_b \leftarrow p_{\theta_t}(\boldsymbol{z}_a), p_{\theta_t}(\boldsymbol{z}_b)$

**23**      $\hat{\boldsymbol{x}}_a', \hat{\boldsymbol{x}}_b' \leftarrow p_{\theta_t}(\boldsymbol{z}_a'), p_{\theta_t}(\boldsymbol{z}_b')$

**24**      Update $\phi_t, \theta_t$ through gradient decent of $\mathcal{L}_{overall}$

**25** **end**

---

### 5.1.2 TrafficSign dataset

We create a traffic sign dataset for further assessing the disentanglement and robustness of the proposed Filter-VAE. Adversarial attacks and defenses in traffic sign recognition (TSR) have been widely investigated, as they are critical to autonomous driving. However, current benchmark datasets like German Traffic Sign Recognition Benchmark (GTSRB) (Stallkamp et al., 2011) and TsinghuaTencent 100K (TT100K) (Zhu et al., 2016) lack factor labels like traffic sign shape, thus they cannot be directly applied to disentangled representation learning. Inspired by Sim2Real (Kadian et al., 2020), we build our traffic sign dataset in a simulation environment by controlling different latent factors, such as shape, color, and orientation. Note that we adopt different colors for each traffic sign to facilitate disentangled representation learning. It is also a contribution to our work that leverages simulated data for real-world prediction.

As shown in Figure 2, it contains 8 different traffic sign models with 5 different shapes: stop (octagon), warning (triangle), speed limit (tall rectangle), oneway (long rectangle), deer crossing (rhombus), handicapped crossing (rhombus), left curve (rhombus), workers ahead (rhombus). Each sign's texture has 10 different colors. Each traffic sign model was rendered on a blue-white checkerboard with a plain background. We orient each model around its bottom in the 2D plane with $[-0.2, 0.2]$ radians and the step size of 0.004. The camera takes an image at a fixed viewpoint. The resulting dataset contains $8 \times 10 \times 100 = 8,000$ RGB $128 \times 128$ images. This dataset will be publicly available upon publication of the paper.

### 5.1.3 Data Preparation

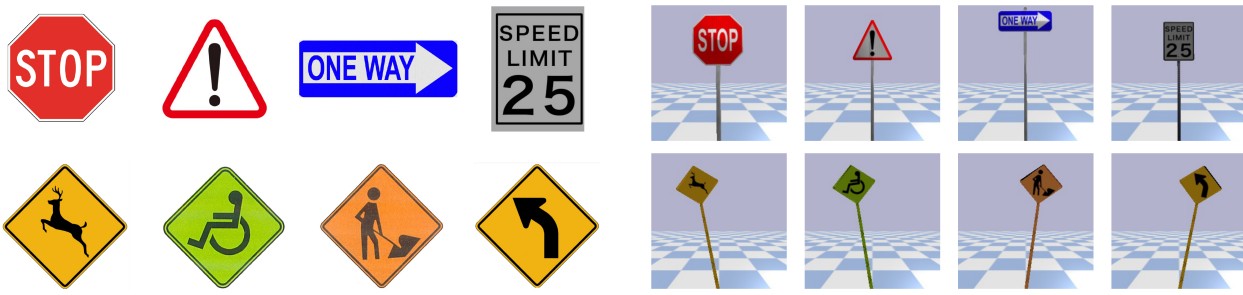

(a) Eight traffic signs in our dataset.      (b) Generated images.

Figure 2: Our synthetic traffic sign dataset. (a) Textures of eight traffic signs cover five different shapes (octagon, triangle, tall rectangle, long rectangle, rhombus). (b) Rendered images in the PyBullet (Greff et al., 2022) simulator.

Table 1: Performance of different methods on the TrafficSign dataset using DCI (Disentanglement), SAP, and MIG, averaged over 5 random seeds. **Bold**: the best method. Underline: the second best method.

| Dataset | TrafficSign | | |
|---|---|---|---|
| Metrics | DCI↑ | SAP↑ | MIG↑ |
| Unsupervised methods | | | |
| $\beta$-VAE$_H$ | 0.239 | 0.042 | 0.177 |
| $\beta$-VAE$_B$ | 0.079 | 0.005 | 0.116 |
| FactorVAE | 0.343 | 0.036 | 0.276 |
| $\beta$-TCVAE | 0.261 | 0.048 | 0.310 |
| ControlVAE | 0.460 | 0.133 | 0.442 |
| Weakly-supervised methods | | | |
| Ada-GVAE | 0.394 | 0.049 | 0.398 |
| Ada-ML-VAE | 0.397 | 0.051 | 0.387 |
| SW-VAE | 0.422 | 0.102 | 0.394 |
| Filter-VAE (Ours) | **0.482** | **0.205** | **0.581** |

In our experiment, we label only a small amount of data for weakly-supervised disentanglement learning. Specifically, we label 10% of the sample pairs $(\boldsymbol{x}_a, \boldsymbol{x}_b)$, where each pair shares identical generative factors except for one distinct factor, as illustrated in Figure 3. This selective labeling facilitates effective latent factor swapping during model training with minimal supervision.

### 5.2 Evaluation on Disentanglement

We evaluate the disentanglement performance of Filter-VAE on four datasets mentioned above, including 3 benchmarks and the TrafficSign dataset.

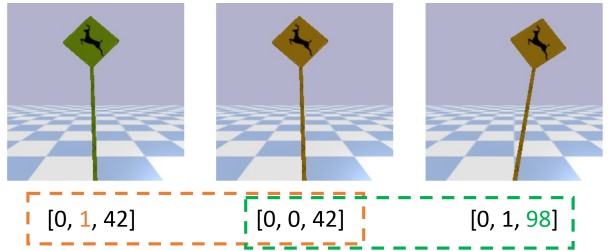

Figure 3: Data samples with three factor labels ([shape, color, orientation]). Samples within each pair only vary in one factor, highlighted with different colors. The dashed box represents a pair of samples.

We compare it with 5 unsupervised baselines: $\beta$-VAE$_H$ (Higgins et al., 2017), $\beta$-VAE$_B$ (Burgess et al., 2018), FactorVAE (Kim & Mnih, 2018), $\beta$-TCVAE (Chen et al., 2018), ControlVAE (Shao et al., 2020b), and 3 weakly-supervised methods: Ada-GVAE, Ada-ML-VAE (Locatello et al., 2020b), SW-VAE (Zhu et al., 2023).

For disentanglement performance evaluation, we use commonly used metrics, DCI-Disentanglement (Eastwood & Williams, 2018), Separated Attribute Predictability (SAP) (Kumar et al., 2018), and Mutual Infor-

Table 2: Performance of different methods on three benchmarks using DCI (Disentanglement), SAP, and MIG, averaged over 5 random seeds. **Bold**: the best method. Underline: the second best method.

| Dataset | 3dChairs | | | 3dShapes | | | dSprites | | |
|---|---|---|---|---|---|---|---|---|---|
| Metrics | DCI↑ | SAP↑ | MIG↑ | DCI↑ | SAP↑ | MIG↑ | DCI↑ | SAP↑ | MIG↑ |
| Unsupervised methods | | | | | | | | | |
| $\beta$-VAE$_H$ | 0.174 | 0.011 | 0.080 | 0.733 | 0.086 | 0.322 | 0.205 | 0.042 | 0.134 |
| $\beta$-VAE$_B$ | 0.117 | 0.010 | 0.063 | 0.439 | 0.052 | 0.349 | 0.389 | 0.081 | 0.360 |
| FactorVAE | 0.121 | 0.011 | 0.057 | 0.685 | 0.066 | 0.249 | 0.199 | 0.036 | 0.123 |
| $\beta$-TCVAE | 0.181 | 0.016 | 0.084 | **0.893** | 0.107 | 0.481 | 0.399 | 0.076 | 0.255 |
| ControlVAE | 0.162 | 0.090 | 0.138 | 0.815 | 0.143 | 0.513 | 0.531 | 0.067 | **0.494** |
| Weakly-supervised methods | | | | | | | | | |
| Ada-GVAE | 0.090 | 0.007 | 0.040 | 0.542 | 0.049 | 0.519 | 0.410 | 0.067 | 0.344 |
| Ada-ML-VAE | 0.094 | 0.006 | 0.039 | 0.496 | 0.040 | 0.465 | 0.438 | 0.070 | 0.376 |
| SW-VAE | 0.170 | 0.073 | 0.092 | 0.772 | 0.068 | 0.328 | **0.560** | 0.090 | 0.472 |
| Filter-VAE (Ours) | **0.230** | **0.133** | **0.175** | 0.836 | **0.171** | **0.523** | 0.514 | **0.104** | 0.455 |

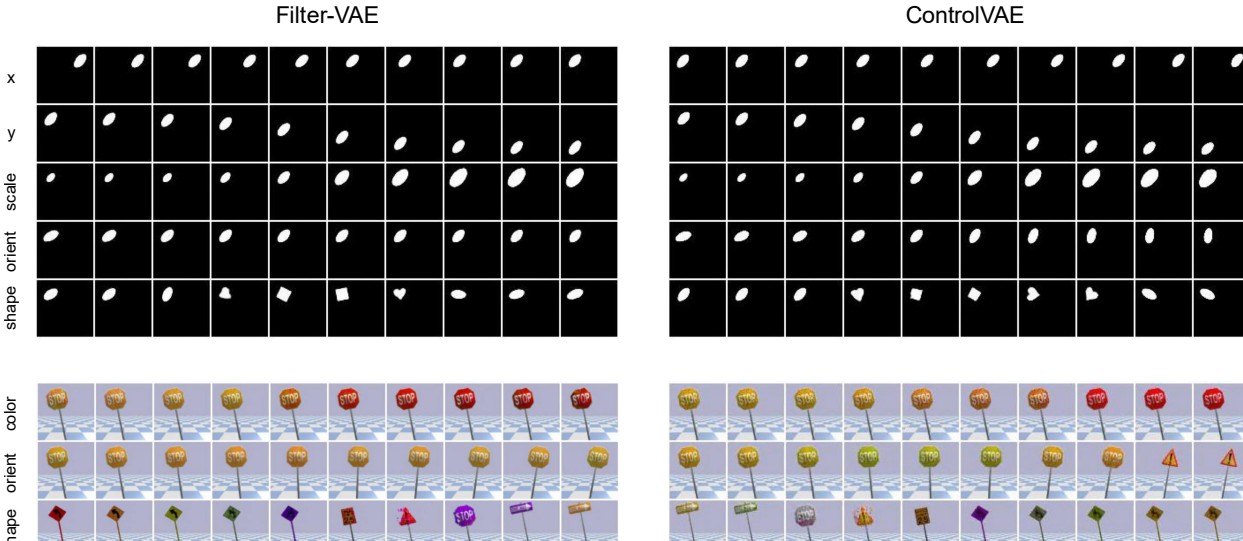

Figure 4: Traverse visualization of top 2 methods on dSprites and TrafficSign datasets. Each row represents one latent feature. **dSprites**: traverse in range of $[-3, 3]$ with step$= 2/3$. Filter-VAE and ControlVAE successfully disentangle all five factors; however, the orientation range of Filter-VAE is small. **TrafficSign**: traverse in range of $[-2, 2]$ with step$= 4/9$. Filter-VAE can effectively disentangle color and orientation, whereas ControlVAE does not perform well.

mation Gap (MIG) (Chen et al., 2018). DCI-Disentanglement measures the weighted average entropy of the probability of each latent factor to predict each generative factor. SAP attributes a prediction score to each pair of latent and generative factors, and measures the average difference between the two highest scores for all generative factors. MIG computes the average difference between the top two latent variables with the highest mutual information over all latent factors.

We present the comparison of disentanglement for different methods in Table 1 and Table 2. It can be observed that the proposed Filter-VAE outperforms baselines using 3dChairs and TrafficSign datasets in all three metrics. Additionally, our method consistently ranks among the top 2 in three metrics in 3dShapes experiments. The improvement is attributed to our method's ability to learn stable and relevant latent factors. For the simple dSprites dataset, the proposed method is slightly worse than ControlVAE and SW-VAE. The main reason is that it does not perform well in disentangling the orientation factor, which

is illustrated in the following Figure 4. Overall, we can conclude that the proposed Filter-VAE exhibits superior performance over baselines in three of these four datasets, except for the simplest one.

Additionally, we illustrate the qualitative results for our Filter-VAE and the second-best baseline in Figure 4. For TrafficSign, we can see that the proposed Filter-VAE can better disentangle two latent factors, shape and orientation. For dSprites, it can be observed that both our method and ControlVAE can successfully discover all five latent factors. However, the orientation (in the 4th row) disentangled by Filter-VAE has small ranges, leading to low MIG in Table 2. This is caused by (1) low explicitness of latent representations and (2) overlap between shape and orient features in the dSprites dataset.

Table 3: Classification accuracy over 17 attacks/corruptions on the TrafficSign dataset. **Bold**: The best method.

| attacks/corruptions | Classification Accuracy ↑ | | | |
|---|---|---|---|---|
| | KEMLP(AdvTrain) | KEMLP(DOA) | FSR(Resnet18) | Filter-VAE + SPN (**Ours**) |
| Impulse Noise ($s$=5) | 0.6056 | 0.6225 | **0.7595** | 0.6153 |
| Gaussian Noise ($s$=5) | 0.4906 | 0.6206 | 0.7310 | **0.7689** |
| Shot Noise ($s$=5) | 0.5381 | 0.6252 | 0.7755 | **0.8119** |
| Defocus Blur ($s$=5) | 0.2525 | 0.2894 | 0.6000 | **0.9123** |
| Glass Blur ($s$=5) | 0.5531 | 0.5719 | 0.5815 | **0.7866** |
| Motion Blur ($s$=5) | 0.6131 | 0.6256 | 0.6020 | **0.8449** |
| Zoom Blur ($s$=5) | 0.4956 | **0.6288** | 0.4195 | 0.5006 |
| Brightness ($s$=8) | **0.6175** | 0.6119 | 0.1890 | 0.4478 |
| Snow ($s$=5) | 0.6100 | 0.5813 | 0.6050 | **0.7044** |
| Frost ($s$=5) | 0.3475 | 0.4600 | 0.4345 | **0.5630** |
| Fog ($s$=5) | 0.1500 | 0.2394 | 0.1350 | **0.3963** |
| Contrast ($s$=5) | 0.3544 | **0.5763** | 0.1385 | 0.1885 |
| Elastic ($s$=5) | 0.6063 | 0.6119 | 0.5975 | **0.7145** |
| Pixelate ($s$=7) | 0.4400 | 0.6213 | 0.6205 | **0.8595** |
| JPEG ($s$=5) | 0.6031 | 0.6225 | 0.6240 | **0.8798** |
| Sticker ($5 \times 5$) | 0.0467 | 0.5100 | 0.1245 | **0.5700** |
| $\mathcal{L}_\infty$ ($\epsilon = 8/255$) | 0.6056 | 0.1381 | 0.3960 | **0.8069** |

### 5.3 Evaluation on Adversarial Robustness

Next, we compare the performance of our method against adversarial attacks with two baselines: KEMLP (Gürel et al., 2021) and FSR (Kim et al., 2023).

In the experiments, our method is evaluated to classify traffic signs under 17 adversarial attacks that contain: (1) **Common corruptions** (Hendrycks & Dietterich, 2019): It contains 15 types of corruptions caused by noise, blur, weather, and digital. We follow the same severity settings $s$ as KEMLP. (2) $\mathcal{L}_\infty$ **PGD attack** (Madry et al., 2018): we set $\epsilon = 8/255$ with 40 iterations to construct perturbations that are unnoticeable to humans while still causing incorrect predictions. (3) **Sticker attack**: This attack attaches a sticker to traffic signs. We use the same adversarial sticker patch generated by (Eykholt et al., 2018) with the size of $5 \times 5$. An exhaustive search is conducted to find the best attack location. Additionally, we use classification accuracy to assess model performance.

The classification is carried out using a Sum-Product Network (SPN) (Poon & Domingos, 2011), a probabilistic graphical model structured as a rooted acyclic directed graph (DAG). Compared to standard classification models, such as multilayer perceptrons (MLPs), SPNs offer probabilistic interpretations, enabling reasoning over dependencies among latent factors with a small sacrifice to accuracy. For completeness, we also report classification results using MLPs and linear classifiers in the Appendix C.

Table 3 illustrates the accuracy comparison of different methods. It can be observed that the proposed Filter-VAE remarkably outperforms baselines in most attack scenarios. This is because our method adopts

Table 4: Ablation study of two filters on the TrafficSign dataset. **Bold**: the best method.

| $F_1$ | $F_2$ | DCI ↑ | SAP ↑ | MIG ↑ |
|-------|-------|-------|-------|-------|
| × | × | 0.4532 | 0.1307 | 0.4203 |
| × | ✓ | 0.4587 | 0.1551 | 0.4873 |
| ✓ | × | 0.4697 | 0.1895 | 0.5514 |
| Ours with both | | **0.4873** | **0.2013** | **0.5831** |

generative high-level factors instead of low-level features to detect traffic signs, improving its robustness against adversarial examples. However, it exhibits limited robustness to strong attacks such as contrast and brightness, as these attacks significantly degrade image quality and interfere with high-level factors such as shape and color. Moreover, we leverage a tractable SPN to provide logical reasoning for predictions. Please refer to the graphical structure in Appendix B. In summary, disentangled latent factors generated by Filter-VAE are robust to most adversarial attacks.

We also investigate the adversarial robustness of Filter-VAE on real-world traffic sign images in Appendix F. The results indicate the potential of the proposed method for real-world traffic sign applications.

### 5.4 Ablation Studies

Lastly, we explore the impact of two key components, hyperparameters and input labels on model performance. We first study the influence of two filters: the relevance filter $F_1$ and the adaptive swapping filter $F_2$. Then we investigate the effect of three hyperparameters: $\omega$ in the objective function and two threshold values $\alpha, \gamma$ of two filters. Lastly, we conduct experiments with imperfect supervisions to evaluate the sensitivity to input labels.

#### 5.4.1 Effect of two filters in the smooth swapping.

Table 4 shows the experimental results with or without using two filters in the smooth swapping. We can observe that the disentanglement performance will drop if one or two filters are removed. These empirical results suggest the importance of two filters to achieve stable swapping.

#### 5.4.2 Effect of Hyperparameters

**Effect of $\omega$ in objective.** Figure 6 in Appendix D shows the impact of $\omega$ on disentanglement performance. We increase $\omega$ from 0 to 1 to balance the swapped reconstructions loss and the original VAE objective. It can be observed that $\omega = 0.4$ exhibits the best result. In addition, there are no significant changes for three metrics with $\omega > 0$, which indicates Filter-VAE is not sensitive with $\omega > 0$. Since 0.4 gives the best performance, we set $\omega = 0.4$ for our experiments.

**Effect of $\alpha$ in Filter 1.** We also study the effect of $\alpha$ that separates meaningful latent factors in the latent space. Setting $\alpha = 0$ implies all latent factors are considered irrelevant, thereby disabling the swapping mechanism. As shown in Figure 7 of Appendix D, using $\alpha$ larger than 0 improves disentanglement performance, highlighting the effectiveness of the swapping strategy. Additionally, small $\alpha$ values can better identify relevant latent factors in the early training iterations. This is attributed to the magnitude gap between relevant and irrelevant latent factors. We set $\alpha = 0.2$ in our experiments.

**Effect of $\gamma$ in Filter 2.** Lastly, we investigate the effect of $\gamma$ by varying its value from 0 to 1. As illustrated in Figure 8 of Appendix D, Filter-VAE achieves the highest disentanglement performance at $\gamma = 0.4$. Setting $\gamma = 0$ disables swapping, while $\gamma = 1$ allows unconstrained swapping. Both extreme conditions result in suboptimal performance. Based on these observations, we choose $\gamma = 0.4$ in our experiments.

#### 5.4.3 Performance under imperfect supervision

We further assess Filter-VAE under imperfect supervision to examine its sensitivity to the selection of labeled pairwise data. Here, imperfect supervision refers to cases where a pair of inputs may differ in more than

Table 5: Performance of Filter-VAE on the TrafficSign dataset with one and random distinct generative factors.

| Metrics | DCI↑ | SAP↑ | MIG↑ |
|---|---|---|---|
| Filter-VAE with one distinct factor | 0.487 | 0.201 | 0.583 |
| Filter-VAE with random distinct factors | 0.476 | 0.185 | 0.557 |

one generative factor. As shown in Table 5, the performance of Filter-VAE remains stable even when we generate image pairs with random number of distinct factors, indicating that our method is not sensitive to imperfect supervision. We attribute this robustness to the adaptive swapping filter $F_2$: when multiple latent factors differ, $F_2$ flexibly treats them as either the first or the second condition in Equation 7, thereby preventing any distinct latent factors from being incorrectly swapped.

## 6 Conclusion

In this paper, we proposed Filter-VAE, a weakly supervised variational autoencoder that employs a filter-based adaptive swapping technique to learn disentangled representations using limited supervision. Specifically, we designed two filters, a relevance filter and an adaptive swapping filter, to identify meaningful latent factors and prevent undesirable swapping of unstable, entangled factors. Filter-VAE can achieve effective disentanglement using only a small subset of pairwise data that contains a single distinct generative factor. Extensive experiments on four datasets demonstrated the superiority of Filter-VAE over existing baselines in both disentanglement and adversarial robustness tasks.

## 7 Limitation

While our proposed method achieves strong performance in both disentanglement and adversarial robustness, it relies on a heuristic assumption that high dim-wise KL divergence corresponds to semantically relevant latent factors. However, this assumption is inspired by prior work (Shao et al., 2020b) and currently lacks a theoretical foundation. As part of future work, we plan to develop a formal justification to better support this design choice.

## Acknowledgments

Research reported in this paper was sponsored in part by NSF CPS 2311086, NSF CIRC 716152, NSF RITEL 2506890, and Faculty Research Grant at William & Mary 141446.

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

## A  Model Configurations and Hyperparameter Settings

### A.1  Filter-VAE

In this experiment, our Filter-VAE uses the same encoder and decoder as ControlVAE (Shao et al., 2020b). The model configurations and hyperparameter settings are presented in Table 6. We adopt the Adam optimizer with $\beta_1 = 0.90$, $\beta_2 = 0.99$ and a learning rate of $10^{-4}$. In addition, following the same settings of ControlVAE, we choose the desired KL-divergence as 19, 18, 10 for dSprites, 3D Chairs, and 3dShapes datasets. For TrafficSign dataset, the desired KL value is set to 11. The dimension of the latent space $z_{dim}$ is set to 10 for all experiments.

For $\alpha$, $\gamma$, and $\omega$, we tuned these parameters in the TrafficSign dataset and applied values to other datasets. $\alpha$ is selected based on the observation that meaningful representation $z_i^+$ has higher dim-wise KL values than others. $\gamma$ is determined by the highest $D_{KL}(q_{\phi,a}(z_i^+)\|q_{\phi,b}(z_i^+))$ among $z_i^+$ in the training process. $\omega$ is the weight to adjust the swap reconstruction loss. Based on the ablation study in Section 5.4, we choose $\omega = 0.4$, $\alpha = 0.2$, $\gamma = 0.4$ in all experiments.

### A.2  SPN

SPN is trained with three Gaussian and one categorical distributions representing class labels with clustering algorithms (Gens & Pedro, 2013; Yang et al., 2022). The result SPN is a valid (complete and decomposable) 7-layer rooted DAG with 240 nodes (13 sum nodes, 77 product nodes, and 150 leaves) and 239 edges. In the trained SPN, we found that splitting color and shape contributes more to the classification because it gives 8 and 9 branches respectively, while orientation only gives 2 branches. It is reasonable because the traffic sign class is mainly determined by its color and shape.

Table 6: Model architectures of Filter-VAE

| Encoder | Decoder |
|---|---|
| Input: $c \times 64 \times 64$ image | Input: $\in \mathbb{R}^{z\_dim}$ |
| $4 \times 4$ conv, stride 2, pad 1. 32 ReLU. | FC 256. 256 ReLU. |
| $4 \times 4$ conv, stride 2, pad 1. 32 ReLU. | $4 \times 4$ transconv. 256 ReLU. |
| $4 \times 4$ conv, stride 2, pad 1. 64 ReLU. | $4 \times 4$ transconv, stride 2, pad 1. 64 ReLU. |
| $4 \times 4$ conv, stride 2, pad 1. 64 ReLU. | $4 \times 4$ transconv, stride 2, pad 1. 64 ReLU. |
| $4 \times 4$ conv, stride 1, pad 1. 256 ReLU. | $4 \times 4$ transconv, stride 2, pad 1. 32 ReLU. |
| FC 256. | $4 \times 4$ transconv, stride 2, pad 1. 32 ReLU. |
| FC $z\_dim \times 2$. | Output: $c \times 64 \times 64$ |

## B  SPN Visualization

Following the idea of the prior work (Yang et al., 2022), we draw a knowledge graph to showcase the logical inference process of SPN for classification in Figure 5. As an instance of an adversary scenario, if the texture of "workers_ahead" is covered with stickers, SPN can still distinguish it from other signs through its color and shape.

## C  Evaluation with Different Classifier

We conduct traffic sign classification using three different backbone models as the classifier: a single-layer linear classifier, a multilayer perception (MLP), and a sum-product network (SPN). Each model is provided

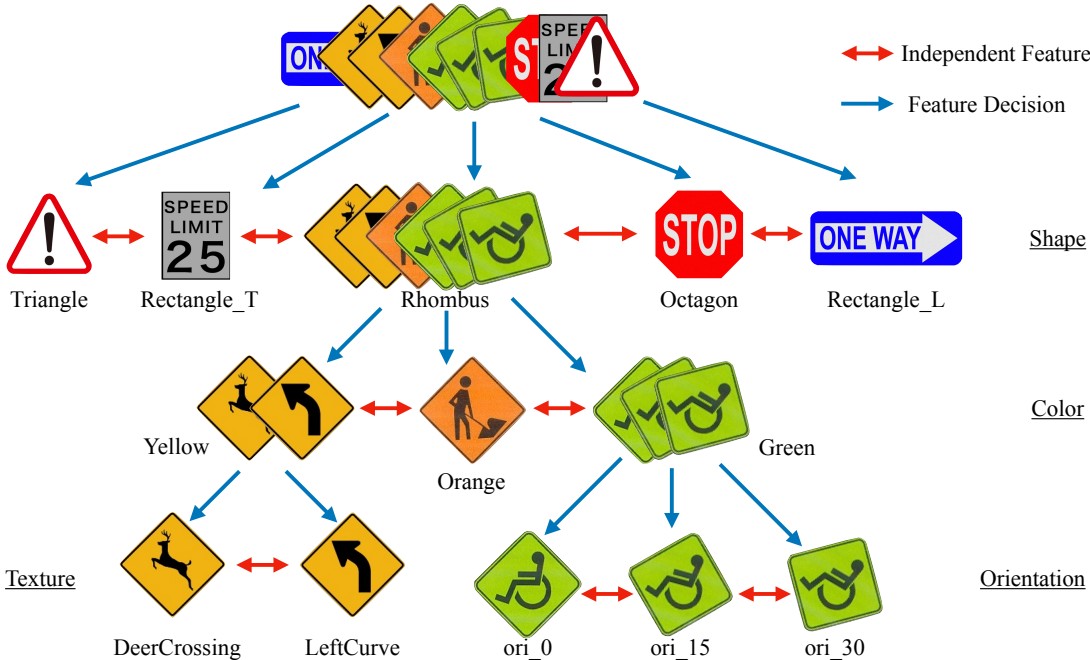

Figure 5: Knowledge graph of the TrafficSign dataset. The red arrow means independent features between data samples. The blue arrow represents the logical feature decision process.

with ground-truth traffic sign labels along with 3 relevant latent factor values corresponding to shape, orientation, and color.

The results are shown in Table 7. Both the MLP and SPN outperform the linear classifier, as they can capture complex interactions among latent factors. Although the SPN achieves slightly lower accuracy than the MLP, it provides tractable decision-making and allows explicit reasoning about how each latent factor contributes to the final prediction. These findings suggest that robustness primarily stems from the disentangled representations rather than the specific choice of classifier.

## D    Effect of Important Parameters

In this section, we illustrate the impact of three hyperparameters $\omega$, $\alpha$, and $\gamma$ on disentanglement performance using the TrafficSign dataset.

### D.1    Effect of $\omega$ in the objective function.

$\omega$ is the weight to balance pairwise ControlVAE loss and swapping reconstruction loss in Equation 10. To investigate its effect, we increase $\omega$ from 0 to 1. As shown in Figure 6, Filter-VAE achieves the best performance in three matrices when $\omega = 0.4$.

### D.2    Effect of $\alpha$ in relevant latent filter.

$\alpha$ is the threshold value to filter out irrelevant latent codes in the relevant latent filter $F_1$. As shown in Figure 7, we investigate the effect of $\alpha$ in the range from 0 to 1. When $\alpha = 0$, all latent dimensions are treated as irrelevant, and no latent code is swapped. Setting $\alpha > 0$ yields improved results, indicating that swapping enhances disentanglement performance. The best performance is achieved at $\alpha = 0.2$. Because meaningless latent codes typically have smaller magnitudes than meaningful ones, a small $\alpha$ value allows the filter to identify relevant latent codes more effectively. In our experiments, we set $\alpha = 0.2$.

Table 7: Classification accuracy over 17 attacks/corruptions on TrafficSign dataset with different classification backbones. **Bold**: the best method.

| attacks/corruptions | Classification Accuracy ↑ | | |
|---|---|---|---|
| | Filter-VAE + Linear | Filter-VAE + MLP | Filter-VAE + SPN |
| Impulse Noise ($s$=5) | 0.5051 | **0.6429** | 0.6153 |
| Gaussian Noise ($s$=5) | 0.6599 | **0.8310** | 0.7689 |
| Shot Noise ($s$=5) | 0.6791 | **0.8606** | 0.8119 |
| Defocus Blur ($s$=5) | 0.7399 | **0.9294** | 0.9123 |
| Glass Blur ($s$=5) | 0.6709 | **0.8314** | 0.7866 |
| Motion Blur ($s$=5) | 0.7003 | **0.8829** | 0.8449 |
| Zoom Blur ($s$=5) | 0.4375 | **0.5153** | 0.5006 |
| Brightness ($s$=8) | 0.3778 | **0.4573** | 0.4478 |
| Snow ($s$=5) | 0.5950 | **0.7748** | 0.7044 |
| Frost ($s$=5) | 0.4579 | **0.6145** | 0.5630 |
| Fog ($s$=5) | 0.3405 | **0.4109** | 0.3963 |
| Contrast ($s$=5) | 0.1250 | **0.1955** | 0.1885 |
| Elastic ($s$=5) | 0.6183 | **0.7655** | 0.7145 |
| Pixelate ($s$=7) | 0.7115 | **0.9008** | 0.8595 |
| JPEG ($s$=5) | 0.7119 | **0.9046** | 0.8798 |
| Sticker ($5 \times 5$) | 0.3650 | 0.5400 | **0.5700** |
| $\mathcal{L}_\infty$ ($\epsilon = 8/255$) | 0.6250 | **0.8175** | 0.8069 |

### D.3 Effect of $\gamma$ in adaptive swapping filter.

$\gamma$ is the upper bound used to determine the stability of latent codes in the adaptive swapping filter $F_2$. Its effect is evaluated by varying from 0 to 1, as shown in Figure 8. Filter-VAE achieves the best disentanglement performance when $\gamma = 0.4$. When $\gamma = 0$, all latent codes are considered unstable, and no swapping is performed. Conversely, when $\gamma = 1$, all relevant codes are unconstrained by the upper bound, which means the training stability is ignored. We can see that both extreme conditions lead to degraded performance. Based on these observations, we set $\gamma = 0.4$ in our experiments.

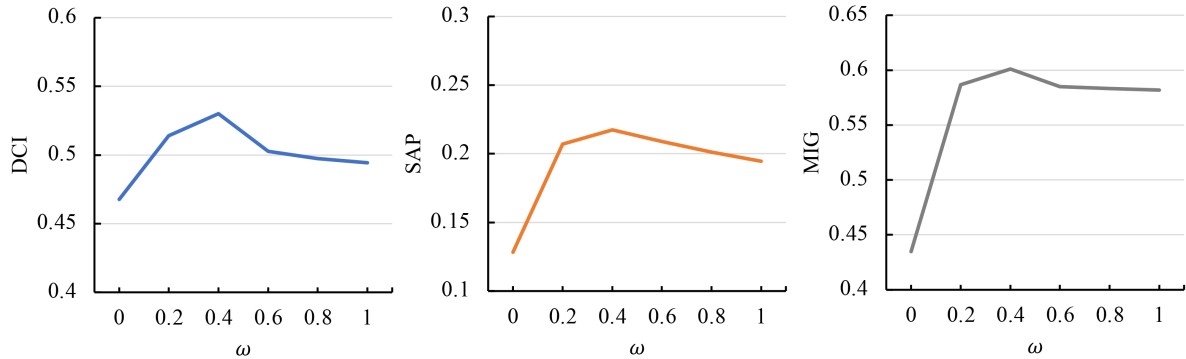

Figure 6: Effect of $\omega$ on disentanglement performance using TrafficSign dataset. We can see that our method is not sensitive with $\omega > 0$.

## E Latent Traversals Visualization

We illustrate the latent traversals of our method and the baselines in Figures. 9, 10, and 11. The baseline methods like Ada-GVAE, Ada-ML-VAE, and $\beta$-VAE$_B$ fail to trade off the reconstruction error and disentanglement, which results in blurry reconstructed images.

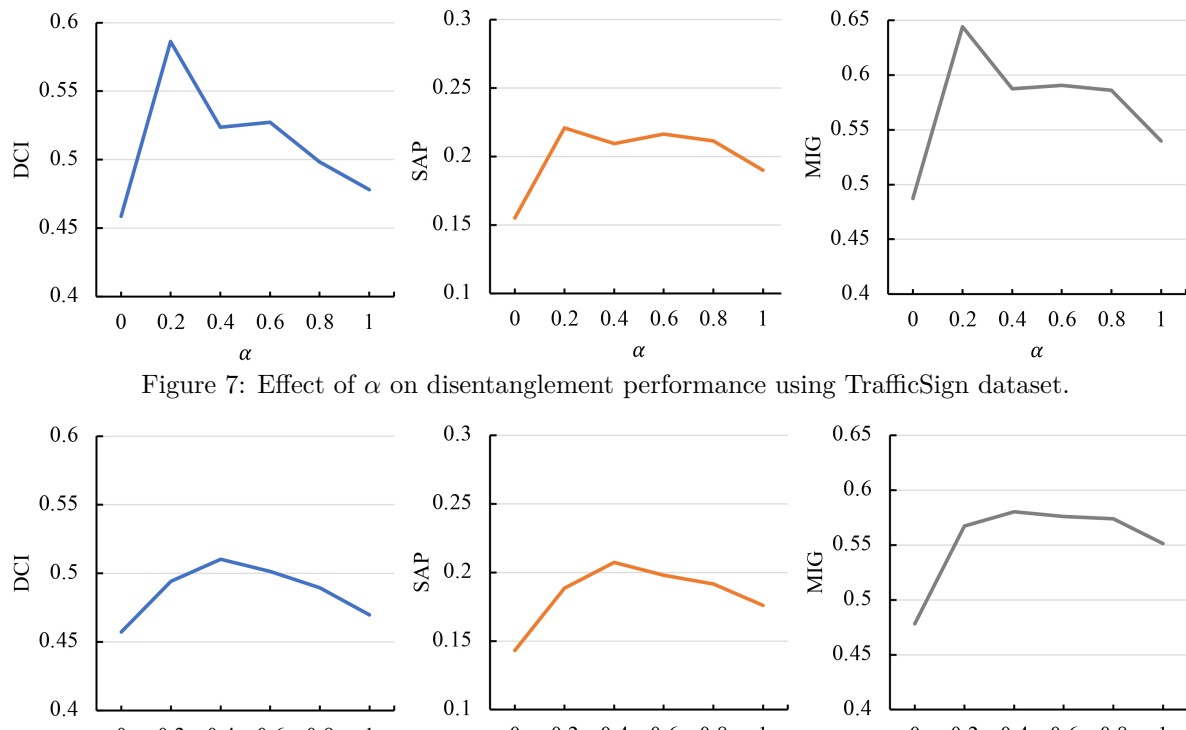

Figure 7: Effect of $\alpha$ on disentanglement performance using TrafficSign dataset.

Figure 8: Effect of $\gamma$ on disentanglement performance using TrafficSign dataset.

### E.1 TrafficSign

Filter-VAE can effectively disentangle color and orientation. Other baselines, except for $\beta$-TCVAE, cannot learn these two factors well. For instance, ControlVAE entangles all features together in the 2nd row. Though $\beta$-TCVAE can also learn both color and orientation, it suffers from poor reconstruction in the dSprites dataset. In addition, separating color from shape remains challenging for all methods when data contains complex texture information.

### E.2 3dShapes

Filter-VAE can effectively disentangle all factors except for object_hue (the 3-rd row), which is entangled with slight shape changes. Though $\beta$-TCVAE achieves a good disentanglement result in 3dShapes as indicated in Table 2, it sacrifices reconstruction quality to get better disentanglement, especially in Figure 11.

### E.3 dSprites

Filter-VAE achieves better disentanglement performance than most baselines, except that the orientation is slightly worse compared to ControlVAE. However, ControlVAE does not perform well in disentangling orientation on the TrafficSign dataset. Moreover, it can only disentangle shape and wall_hue in the 3dShapes dataset.

## F  Real-world Experiments

The effectiveness of the proposed method on real-world data is crucial for practical deployment. However, a significant limitation lies in the absence of real-world traffic sign benchmarks with controlled latent factors. To address this issue, we leverage In-Context Learning (ICL) with the state-of-the-art vision-language model Gemini Nano Banana to construct a dataset of 1,960 real-world traffic sign images. Specifically, we use

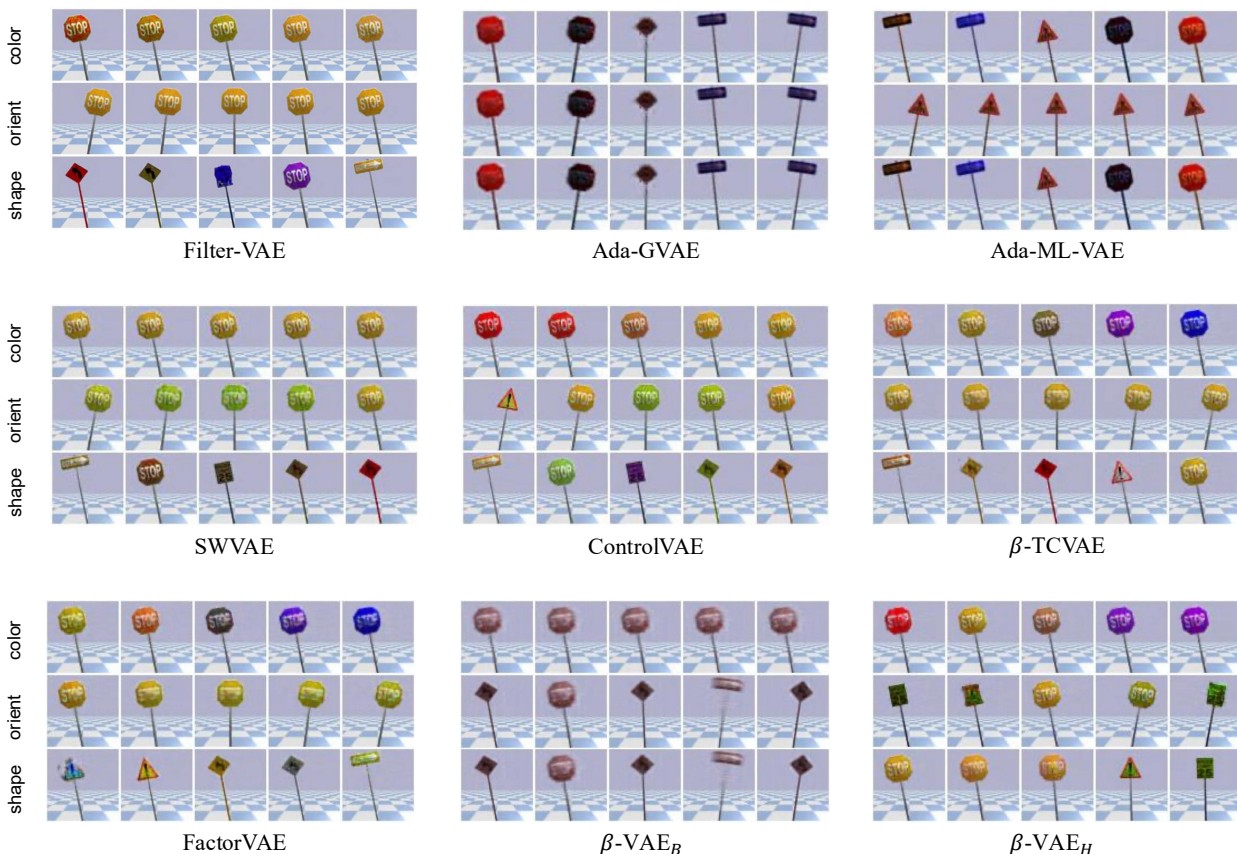

Figure 9: Latent traversals on the TrafficSign dataset. Each row represents one latent factor. Traverse in the range of $[-2, 2]$ with step $= 1$. Filter-VAE and $\beta$-TCVAE can disentangle color and orientation successfully. However, the reconstruction quality of $\beta$-TCVAE is worse than Filter-VAE.

publicly available traffic sign images as ICL examples and design prompts to generate images across eight traffic sign categories under 35 distinct real-world driving scenarios. Each scenario includes seven randomized rotations and eight color variations, resulting in a total of $35 \times 7 \times 8 = 1,960$ images. The latent factors (color, orientation) are explicitly controlled via prompt engineering. Notably, due to Gemini's limitation in executing precise rotation degrees from prompt specifications, each traffic sign is randomly rotated seven times to approximate orientation diversity. Details of the dataset and the generation pipeline are provided in the supplementary material.

We present classification results in Table 8. Filter-VAE achieves either the best or second-best performance under most attacks and corruptions, with exceptions of Frost and Gaussian Noise. Nonetheless, its performance under these two corruptions remains competitive, exhibiting only minor drops of 0.0750 and 0.0276 respectively, compared to the best results. These findings demonstrate the strong potential of our approach for real-world traffic sign applications.

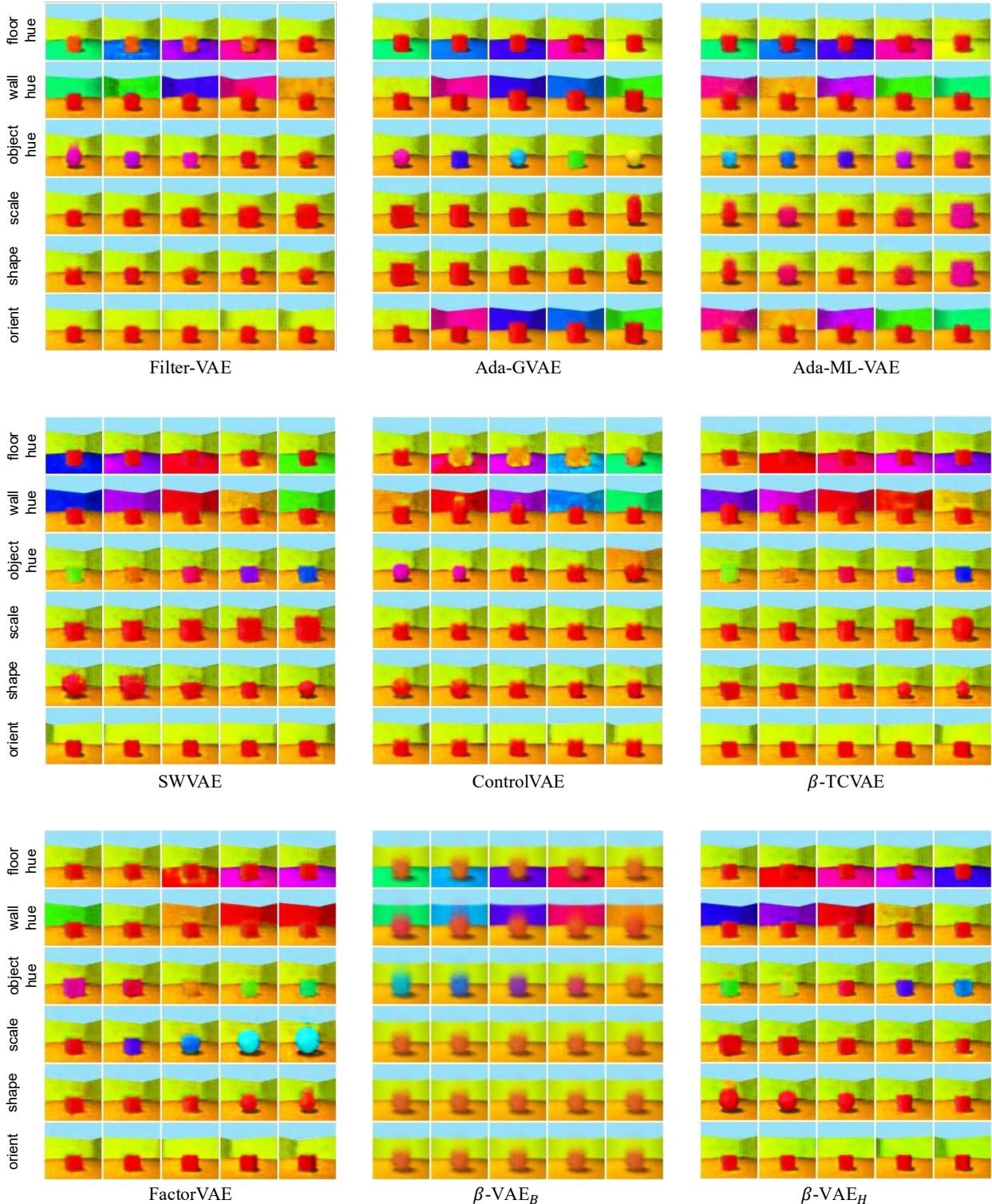

Figure 10: Latent traversals on the 3dShapes dataset. Each row represents one latent factor. Traverse in the range of $[-1, 1]$ with step $= 1/2$. Filter-VAE can successfully disentangle all features except for a slight shape change in object_hue.

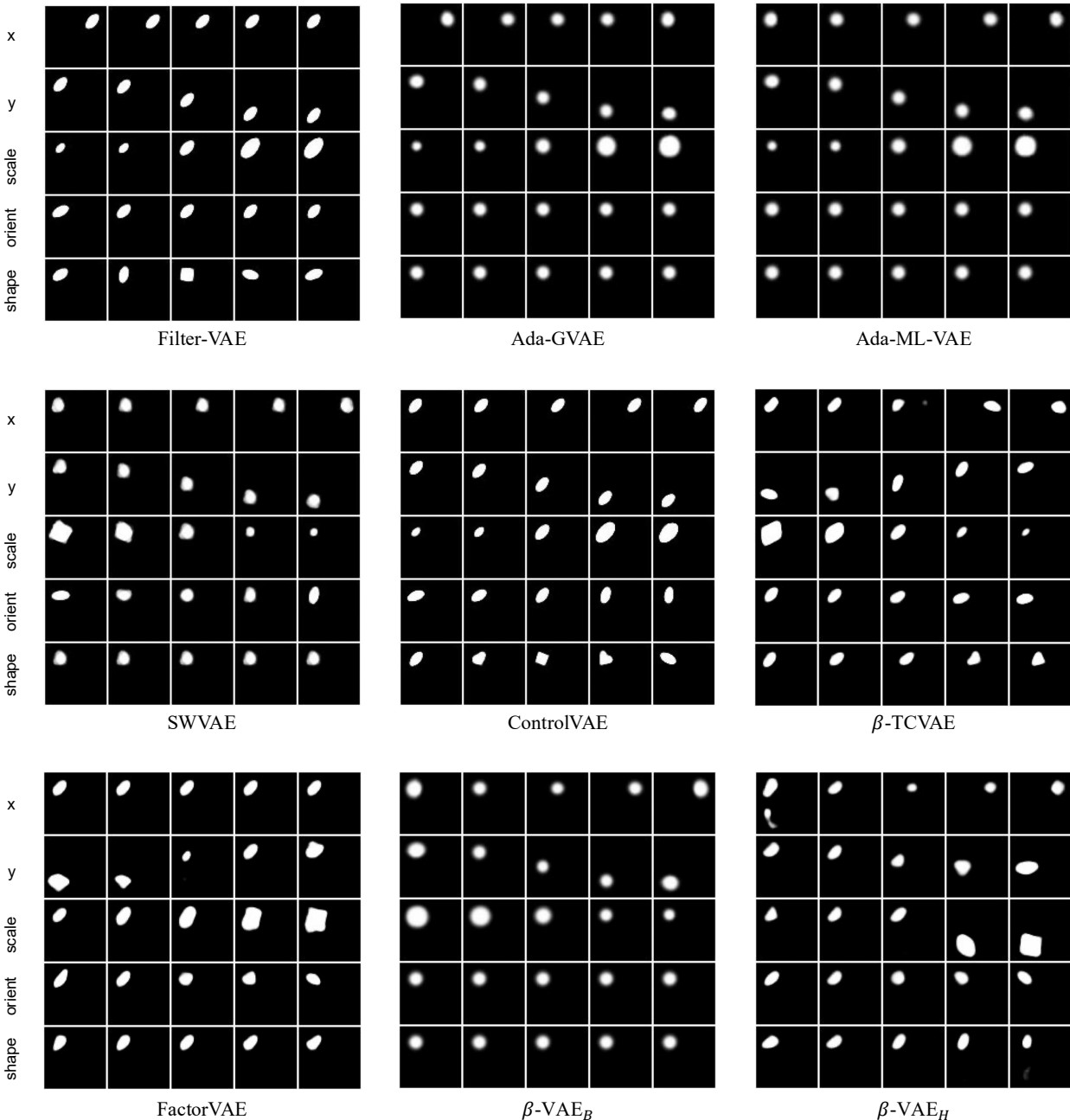

Figure 11: Latent traversals on the dSprites dataset. Each row represents one latent factor. Traverse in the range of $[-3, 3]$ with step $= 3/2$. We can observe that Filter-VAE and ControlVAE can successfully disentangle five latent factors, while the other baselines cannot.

Table 8: Classification accuracy over 17 attacks/corruptions on the real-world traffic sign dataset. **Bold**: The best method. Underline: The second best method.

| attacks/corruptions | Classification Accuracy ↑ | | | |
|---|---|---|---|---|
| | KEMLP(AdvTrain) | KEMLP(DOA) | FSR(Resnet18) | Filter-VAE + SPN (**Ours**) |
| Impulse Noise ($s$=5) | **0.9245** | 0.8020 | 0.8102 | 0.8704 |
| Gaussian Noise ($s$=5) | 0.9245 | 0.8265 | **0.9490** | 0.9214 |
| Shot Noise ($s$=5) | 0.9245 | 0.8265 | **0.9347** | 0.9281 |
| Defocus Blur ($s$=5) | 0.9143 | 0.8184 | 0.8878 | **0.9352** |
| Glass Blur ($s$=5) | 0.8939 | 0.8020 | 0.8939 | **0.9306** |
| Motion Blur ($s$=5) | 0.8816 | 0.7918 | 0.8286 | **0.9332** |
| Zoom Blur ($s$=5) | 0.8224 | 0.7735 | 0.8388 | **0.8643** |
| Brightness ($s$=8) | 0.2163 | 0.1755 | **0.2204** | 0.2189 |
| Snow ($s$=5) | 0.5082 | **0.5755** | 0.3776 | 0.5724 |
| Frost ($s$=5) | 0.4551 | **0.4959** | 0.4776 | 0.4209 |
| Fog ($s$=5) | **0.3265** | 0.3000 | 0.1776 | 0.3133 |
| Contrast ($s$=5) | 0.2714 | 0.2816 | 0.2122 | **0.2862** |
| Elastic ($s$=5) | 0.8959 | 0.7980 | 0.8857 | **0.9112** |
| Pixelate ($s$=7) | 0.9122 | 0.8163 | 0.9020 | **0.9357** |
| JPEG ($s$=5) | 0.9245 | 0.8224 | 0.9327 | **0.9378** |
| Sticker ($5 \times 5$) | 0.6694 | **0.9082** | 0.2633 | 0.8952 |
| $\mathcal{L}_\infty$ ($\epsilon = 8/255$) | **0.9265** | 0.6122 | **0.9265** | 0.9071 |

