# OpenReview forum: "Weakly-Supervised Disentangled Representation Learning via Filter-Based Adaptive Swapping"
_TMLR — Accepted by TMLR_

### Review · Reviewer_fqeo · 2025-10-23

**Summary Of Contributions:**

This paper introduces Filter-VAE, a weakly supervised model that improves disentangled representation learning. It uses two filters: one to remove meaningless latent factors and another to align stable ones,  allowing more reliable disentanglement with minimal labeled data. The authors also build a synthetic traffic sign dataset to evaluate both disentanglement quality and adversarial robustness. Experiments show that Filter-VAE outperforms existing methods on several benchmarks while using less supervision, and it also notably improves model robustness against a wide range of adversarial attacks.

Strength:
1. Proposed a new algorithm, VAE with two filters.
2. Developed a synthetic traffic sign dataset.

Weakness:
1. The overall presentation feels somewhat rushed and underdeveloped.
2. Did not justify reproducibility.
3. Experiment results (Table 1&2) are based on only 3 replications.

**Audience:**

Yes

**Audience Explanation:**

Yes, the topic of weakly supervised disentanglement and model robustness is relevant to TMLR readers, although the current version may need stronger presentation and empirical support to attract wider attention.

**Broader Impact Concerns:**

I did not find any concerns.

**Claims And Evidence:**

Yes

**Claims Explanation:**

If the authors could justify reproducibility, and explain why there are only three replications, the results should be convincing.

**Requested Changes:**

See weaknesses.
1. Presentation. The authors should carefully revise their manuscript, including but not limited to:

1.a. Overall Organization: The main text often feels loosely structured: key algorithmic components are deferred to the supplementary material, while large figures and tables take excessive space without conveying much additional insight.

1.b. Problem Statement: I'm quite confused. Did you omit the dimension of each data point x_i, or in your mapping, f(x) -> z, x indeed contains N data points and each data point is a scalar? If x_i (i=1,2,...,N) is a vector, you should use bold x_i, and "Given a batch of N observed data" (sample-level) and mapping (model-level) should not be in same sentence. If x in "f(x) -> z" indeed contains N data points  and each data point is a scalar, what does it mean that using N observations to learn one latent factor?

1.c. Math notations: some text in math looks not nice, e.g., ConVAE in equation (2), Diff in equation (7); in the last paragraph of Page 5, Gaussian distribution N(mu, sigma^2), why does here (mu, sigma^2) not contain any subscript or superscript; in equation (4), q should follow (using "~" in math) instead of equals to ("=") normal distribution.

1.d. Wording: In section 5.1.2, "stop (octagon), warning (triangle), speed_limit (rectangle_tall), oneway (rectangle_long), deer_crossing (rhombus), handicapped_crossing (rhombus), left_curve (rhombus), workers_ahead (rhombus).", the underscore is not the scientific presentation stype.

1.e. Citation: almost all of the parenthesis in citations are missing, e.g., "DRL has demonstrated success in several domains such as image generation, natural language processing, and recommendation systems Wang et al. (2024)."

2. Reproducibility: please (anonymously) provide (at least part of) the codes to justify reproducibility, especially for the applied work.

3. Results justifiication: Please explain why there are only three replications/seeds.

---

> ### Author Response · Authors · 2025-11-21
>
> ## Q1.a: Improve the overall organization.
> **A1.a**: Per your suggestion, we have moved the key algorithmic description from the supplementary material to the main paper for better readability and completeness. Additionally, three ablation figures have been relocated to Appendix D to optimize space and improve the overall presentation.
>
> ## Q1.b: Confusion about the dimension of each data point x_i and the meaning of "using N observations to learn one latent factor".
> **A1.b**: Thanks for your suggestion. We have changed $x_i$ to bold symbol $\boldsymbol{x}_i$, which corresponds to an input image. We use the bold symbol $\boldsymbol{z}$ to denote the latent representation, which lies in a $d$-dimensional latent space. Accordingly, the mapping $f(\boldsymbol{x})\rightarrow \boldsymbol{z}$ represents the process of learning $d$ latent factors that characterize the underlying features of the input images $\boldsymbol{x}$.
>
> ## Q1.c: Update math notations in: ConVAE in equation (2); Diff in equation (7); Gaussian distribution N(mu, sigma^2) in the last paragraph of Page 5; q should follow (using "~" in math) in equation (4).
>
> **A1.c**: Per your suggestions, we have changed $L_{ConVAE}$ to $L_{control}$ in Equation (2), and $Diff(z_i^{+})$ to $D(z_i^{+})$ in Equation (7) to make notations look better. We have added a subscript $i$ to the Gaussian distribution on page 5. We have changed "=" to "$\sim$" in Equation 4.
>
> ## Q1.d: The underscore is not the scientific presentation type in section 5.1.2.
> **A1.d**: We have removed the underscore in Section 5.1.2.
>
> ## Q1.e: The parentheses in citations are missing.
> **A1.e**: Thanks for your suggestion. We have added parentheses to in-text citations.
>
> ## Q2: Please (anonymously) provide the codes to justify reproducibility, especially for the applied work.
> **A2**: We have uploaded the source code as supplemental materials for reproduction.
>
> ## Q3: Please explain why there are only three replications/seeds.
> **A3**: The variability of our experimental results is relatively low, which means that changing random seeds does not lead to significant performance fluctuations. Therefore, following the practice in prior work [1], we report results averaged over three random seeds, which we found sufficient to ensure stable and representative outcomes.
>
> > [1] Tim Z. Xiao, et al. A Note on Generalization in Variational Autoencoders: How Effective Is Synthetic Data and Overparameterization? TMLR 2025.

---

> > ### Comment · Reviewer_fqeo · 2025-12-27
> > **Thank authors for the response**
> >
> > Thank authors for the response, the writeup is much better. And you almost solve all my questions, except for "3 seeds". From the statistic perspective, it is hard to say "The variability of our experimental results is relatively low" based on only replications. Btw, you may also modify the write-up in your key algorithm  (eg. Diff in Algorithm 1).

---

> ### Author Response · Authors · 2025-12-27
>
> We have updated Tables 1 and 2 to include results averaged over five random seeds. The outcomes remain consistent with our previous findings. In addition, we have further refined the write-up of Algorithm 1, including the description of the Diff operation.

---

### Review · Reviewer_f7K6 · 2025-10-23

**Summary Of Contributions:**

In the paper, the author presents Filter-VAE, a weakly supervised VAE using a filter-based adaptive swapping strategy for stable, meaningful disentanglement. The paper introduces two key filters: a "Relevance filter" that removes semantically meaningless latent factors, and an "Adaptive swapping filter" that exchanges latent factors that have reached stability during training. This dual-filter mechanism enables Filter-VAE to adaptively swap only stable and semantically aligned latent factors, resulting in robust and meaningful representations.
The author also presents comparable results with other weakly supervised methods on the traffic sign dataset. And the results look promising.
Although the method appears promising compared to others, the dataset seems somewhat limited as it’s synthetic, allowing the authors significant control over the attributes of the traffic dataset, including the shapes and colors of traffic signs.

The paper states that Filter-VAE targets disentangled representation learning for traffic sign recognition (TSR), noting that existing benchmarks, such as GTSRB and TT100K, lack factor labels, including sign shape. However, the paper doesn’t clearly explain how this approach could be applied to real-world traffic scenes, where such controlled conditions don’t exist.

**Audience:**

Yes

**Audience Explanation:**

Yes, Developers working on diffusion models, representation learning, and robust perception for autonomous driving (not limited to) would benefit most.

**Claims And Evidence:**

Yes

**Claims Explanation:**

Yes, the paper shows a comparison of results with other relevant algorithms.

**Requested Changes:**

The paper would be even stronger if it included benchmarks on real-world data, where conditions are less controlled. Even if the results differ from those on synthetic datasets, such experiments would provide valuable insights into real-world performance and help future developers understand practical challenges and key takeaways.

---

> ### Author Response · Authors · 2025-11-21
>
> ## Q1: The paper would be stronger if it included real-world datasets.
>
> **A1**: Following your suggestion, we further evaluated our method using real-world traffic sign data. Specifically, we collected real-world traffic sign images from Google Images and employed Gemini Nano Banana for data augmentation, generating 1,960 samples with diverse shapes, orientations, and colors. The link to this dataset is provided in the supplementary materials. Using the same training and testing pipelines as in our adversarial classification experiments, we report the classification results in Table 1. We can see that our method achieves the best or second-best adversarial accuracy across most corruption types, except for Frost and Gaussian Noise. Nonetheless, its performance under these two corruptions remains competitive, exhibiting only minor drops of 0.0750 and 0.0276, respectively, compared to the best results. These findings demonstrate the strong potential of our approach for real-world traffic sign applications.
>
> Table 1. Real-world traffic sign classification results. **Bold**: best method.
>
> | Attacks/Corruptions  | KEMLP (AdvTrain) | KEMLP (DOA) | FSR (ResNet18) | Filter-VAE + SPN (**Ours**) |
> | -------------------- | ---------------- | ----------- | -------------- | --------------------------- |
> | Impulse Noise (s=5)  | **0.9245**      | 0.8020      | 0.8102         | 0.8704                      |
> | Gaussian Noise (s=5) | 0.9245           | 0.8265      | **0.9490**    | 0.9214                      |
> | Shot Noise (s=5)     | 0.9245           | 0.8265      | **0.9347**    | 0.9281                      |
> | Defocus Blur (s=5)   | 0.9143           | 0.8184      | 0.8878         | **0.9352**                      |
> | Glass Blur (s=5)     | 0.8939           | 0.8020      | 0.8939         | **0.9306**                      |
> | Motion Blur (s=5)    | 0.8816           | 0.7918      | 0.8286         | **0.9332**                      |
> | Zoom Blur (s=5)      | 0.8224           | 0.7735      | 0.8388         | **0.8643**                      |
> | Brightness (s=8)     | 0.2163           | 0.1755      | **0.2204**    | 0.2189                  |
> | Snow (s=5)           | 0.5082           | **0.5755** | 0.3776         | 0.5724                      |
> | Frost (s=5)          | 0.4551           | **0.4959** | 0.4776         | 0.4209                      |
> | Fog (s=5)            | **0.3265**      | 0.3000      | 0.1776         | 0.3133                      |
> | Contrast (s=5)       | 0.2714           | 0.2816      | 0.2122         | **0.2862**                      |
> | Elastic (s=5)        | 0.8959           | 0.7980      | 0.8857         | **0.9112**                      |
> | Pixelate (s=7)       | 0.9122           | 0.8163      | 0.9020         | **0.9357**                      |
> | JPEG (s=5)           | 0.9245           | 0.8224      | 0.9327         | **0.9378**                      |
> | Sticker (5×5)        | 0.5919           | **0.9082** | 0.2633         | 0.8952
> | $\mathcal{L}_\infty$ ($\epsilon$ = 8/255)      | **0.9265**       | 0.6122      | **0.9265**         | 0.9071                      |

---

> > ### Comment · Reviewer_f7K6 · 2025-12-16
> > **Benchmarks look good.**
> >
> > The benchmarks look good. Thanks for sharing.

---

### Review · Reviewer_3nbZ · 2025-12-12

**Summary Of Contributions:**

Summary:

This paper proposes Filter-VAE, a weakly supervised vae designed to learn disentangled representations using limited pairwise supervision. Building on prior work in latent factor swapping (e.g., SW-VAE), the key contribution is a filter-based adaptive swapping strategy that aims to improve training stability and semantic alignment of latent factors. Specifically, the method introduces: (1) a relevance filter, which removes latent dimensions that is semantically meaningfulless based on the KL divergence. (2) an adaptive swapping filter, which swaps only latent factors that are both relevant and sufficiently stable, as measured by pairwise KL between latent posteriors. The method is evaluated on three standard disentanglement dataset and a newly introduced synthetic dataset. The results indicate that it generally outperforms unsupervised and weakly supervised baselines, particularly on more complex datasets, while using relatively weak supervision.

Strengths:

Clear and Intuitive High-Level Idea: The paper addresses a well-known issue in weakly supervised disentanglement: unstable or premature latent swapping, by introducing a principled gating mechanism. The use of KL divergence both to identify meaningful latent dimensions and to assess their stability is conceptually simple and well-motivated.
Incremental but Solid Methodological Contribution: While the approach builds directly on prior latent-swapping VAEs, the introduction of two explicit filters (relevance and stability) represents a meaningful refinement over SW-VAE. The method reduces reliance on strong assumptions (e.g., warm-up stages or precise annotations of varying factors), which is aligned with the goals of weak supervision.

Weaknesses:

1. Although sensitivity analyses are provided, the method still depends on manually chosen thresholds that are tuned on a specific dataset and then transferred to others. This raises direct questions about robustness across domains with very different latent distributions.
2. On dSprites, Filter-VAE does not consistently outperform strong baselines such as ControlVAE, particularly on orientation factors.
3. The robustness gains are demonstrated using an SPN classifier trained on disentangled representations. While the results are strong, it is not entirely clear how much of the improvement is attributable to disentanglement itself versus the choice of classifier or the synthetic nature of the dataset.
4. Both the relevance filter and the adaptive swapping filter rely on dim-wise KL divergence as a proxy for semantic meaningfulness and stability. While empirically motivated, this choice is heuristic and lacks a strong theoretical justification. In particular, high KL divergence does not necessarily imply semantic alignment with a single generative factor, especially in entangled or partially disentangled cases.

Overall:

This paper presents a competent and well-executed incremental contribution to weakly supervised disentangled representation learning. The high-level idea is clear, the method is carefully designed, and the experimental evaluation is largely sufficient to support the main claims, particularly regarding stability and reduced supervision. While the novelty is moderate and the reliance on heuristic thresholds is a limitation, the work is solid and relevant to the TMLR audience, especially for readers interested in practical improvements to disentanglement methods rather than fundamentally new theory. Hence I recommend for acceptance.

**Audience:**

Yes

**Audience Explanation:**

The topic is relevant and potentially interest the community.

**Claims And Evidence:**

Yes

**Claims Explanation:**

The claims made in the submission are supported by the designed experiments.

**Requested Changes:**

See Weaknesses Above

---

> ### Author Response · Authors · 2025-12-13
>
> ## Q1: The method requires manually selecting thresholds. Its robustness across domains with very different latent distributions is unknown.
>
> **A1**: In our experiments, we tune threshold values on the TrafficSign dataset and directly apply them to other benchmarks, demonstrating that these values are not sensitive across domains. The two thresholds in our framework regulate the model's learning behavior, rather than capture any dataset-specific features. For more datasets with very different latent distributions, we plan to investigate them further in future work.
>
> ## Q2: Why does Filter-VAE not outperform well on orientation factors on the dSprites dataset?
>
> **A2**: For dSprites, the orientation factor is partially entangled with orientation, as shown in Figure 4. We observe this phenomenon in ControlVAE and other methods as well, indicating that orientation is hard to disentangle perfectly. This is the main reason why Filter-VAE doesn't perform well in the orientation factor. Nevertheless, our method outperforms the baselines on other datasets.
>
> ## Q3: It is not entirely clear how much of the improvement is attributed to disentanglement itself versus the choice of classifier.
>
> **A3**: In Appendix C, we evaluate both SPN and MLP to assess the impact of classifier choice on performance. As shown in Table 7, both the SPN and MLP achieve strong results compared with baselines in Table 3. It suggests that the observed robustness primarily arises from the disentangled representations, rather than from the specific choice of classifier.
>
> ## Q4.1: The heuristic that the dim-wise KL divergence is a proxy for semantic meaningfulness and stability lacks a strong theoretical justification.
>
> **A4.1**: The heuristic is inspired by prior works such as ControlVAE and $\beta$-VAE. For theoretical justification, we plan to explore this in future work, as discussed in Section 7 of our revised version.
>
> ## Q4.2: High KL divergence does not imply semantic alignment with a single generative factor.
>
> **A4.2**: Please note that in our work, high dim-wise KL divergence values are used to filter out irrelevant latent factors, rather than to imply fully disentangled representations. The swapping process and corresponding objectives ensure the disentanglement.

---

### Decision · Action_Editor_aMxS · 2026-01-20

**Recommendation:** Accept as is

**Additional Comments:**

The paper proposes Filter-VAE, a method for weakly supervised disentangled representation learning that employs a relevance filter and an adaptive swapping mechanism. The goal is to improve the stability and semantic alignment of latent factors without requiring full supervision.

Three reviewers evaluated the submission, and all have recommended acceptance. The reviewers highlighted the clear motivation and the technical soundness of the proposed dual-filter strategy. They also appreciated the evaluation on both standard benchmarks and the newly created traffic sign dataset.

Initial concerns regarding the statistical significance of the results (due to a low seed count) and the lack of real-world evaluation were effectively addressed by the authors during the rebuttal phase.  There is a consensus that the work is technically sound, the experiments are convincing, and it represents a clear contribution to the TMLR audience.

**Audience:**

Yes

**Audience Explanation:**

The findings are of interest to researchers working on variational autoencoders, disentangled representation learning, and weakly supervised learning.

**Claims And Evidence:**

Yes

**Claims Explanation:**

The paper proposes a weakly supervised variational autoencoder designed to improve disentangled representation learning. The authors introduce a "Filter-Based Adaptive Swapping" strategy, which utilizes two specific filters: a Relevance Filter to prune semantically meaningless latent factors, and an Adaptive Swapping Filter that exchanges only those factors that have reached stability. The method is evaluated on standard benchmarks as well as a synthetic traffic sign dataset created by the authors. The claims regarding improved disentanglement and adversarial robustness are supported by accurate and clear evidence.